# Addressing Bullying and Cyberbullying in Public Health: A Systematic Review of Interventions for Healthcare and Public Health Professionals

**DOI:** 10.3390/ijerph22111682

**Published:** 2025-11-06

**Authors:** Stephanie F. Dailey, Rosellen R. Roche, Megan C. Sharkey

**Affiliations:** 1Counseling Program, College of Education and Human Development, George Mason University, Fairfax, VA 22030, USA; 2Department of Family Medicine and Population Health, School of Medicine, Virginia Commonwealth University, Richmond, VA 23298, USA; rosellen.roche@vcuhealth.org; 3Educational Psychology Program, College of Education and Human Development, George Mason University, Fairfax, VA 22030, USA; msharke@gmu.edu

**Keywords:** bullying, cyberbullying, public health, healthcare professionals, bullying prevention, SHIELD framework

## Abstract

Bullying and cyberbullying constitute urgent public health challenges, contributing to significant psychological, social, and developmental harms among youth worldwide. While schools have traditionally served as the primary context for prevention, these efforts are often limited in scope, duration, and systemic integration. Healthcare and public health professionals are uniquely positioned to contribute to early identification, prevention, and resilience-building, but their roles are not consistently integrated into bullying prevention frameworks. This systematic review, conducted in accordance with PRISMA 2020 guidelines, synthesized 12 empirical studies published between 2013 and 2023 that examined healthcare- and public health–led interventions addressing bullying and cyberbullying among children and adolescents. Using a narrative synthesis mapped onto the SHIELD framework (Strengths, Healing, Interventions, Empowerment, Learning, Development), six themes emerged: (1) screening and early identification protocols, (2) family and community involvement, (3) variable focus on mental health and well-being, (4) multi-component, school-based interventions, (5) cognitive-behavioral and solution-focused interventions, and (6) online and digital interventions. Findings highlight the potential of health professionals to deliver trauma-informed, empowerment-based, and culturally responsive approaches that extend beyond traditional educational settings. Recommendations emphasize cross-sector collaboration, integration of digital tools, and equity-centered practices to strengthen prevention, intervention, and resilience-building. This review underscores the critical role of healthcare and public health professionals in creating safer, more supportive environments for youth.

## 1. Introduction

Bullying and cyberbullying continue to be significant public health issues due to their widespread prevalence and profound negative effects on the mental, emotional, and social well-being of children and adolescents [1,2,3]. These behaviors, which include physical, verbal, relational, and digital aggression, contribute to anxiety, depression, somatic symptoms, social isolation, and diminished academic and social functioning [4,5,6]. Given the multifaceted nature of bullying, effective prevention and intervention require comprehensive, multidisciplinary approaches involving healthcare and public health professionals, educators, families, and communities [2,7,8]. This systematic review and narrative synthesis integrate evidence from various studies focusing on interventions relevant to healthcare and public health professionals, particularly nurses, educators, and mental health providers, emphasizing effective strategies, challenges, and implications for practice and policy.

Bullying manifests in multiple forms, including verbal, physical, social/relational, and cyberbullying, each displaying distinct yet overlapping characteristics [3]. Bullies often exhibit aggressive, dominant, or impulsive traits, frequently influenced by family environments characterized by inadequate supervision or neglect [9]. Victims commonly experience emotional distress, social isolation, and mental health concerns, such as anxiety, depression, and psychosomatic complaints [1,5,6]. Cyberbullying adds further complexity because its anonymity, persistence, and wide reach often intensify psychological harm [10,11,12]. Healthcare professionals must recognize these manifestations and their effects to effectively tailor prevention and intervention strategies [2,13]. In practice, many health-led programs address both in-person and digital aggression within unified prevention frameworks, reflecting overlapping risk and protective factors as well as delivery modalities [14,15,16].

Cross-national evidence supports this integrated perspective. A study conducted in China found that adolescents’ moral disengagement has been found to positively predict cyberbullying perpetration, with empathy moderating this relationship, suggesting that interventions targeting moral reasoning and empathy may enhance prevention models [17]. In Japan, a meta-analysis of 85 universal, school-based social–emotional learning programs reported significant improvements in social–emotional skills, attitudes, and behavior, highlighting the value of coordinated implementation between schools and families [18]. These findings demonstrate that early detection of bullying behaviors and victimization remains essential to prevent escalation and long-term consequences [1,4,8].

While school-based interventions have long been the cornerstone of bullying prevention efforts, they are often limited in scope, duration, and reach [19,20]. Many programs are curriculum-based or delivered through short-term initiatives, with inconsistent implementation across schools and districts [21]. Furthermore, these interventions frequently prioritize behavior management and disciplinary responses over trauma-informed, preventive strategies that address the root causes of bullying and its long-term effects [22,23]. Although some mental health-focused interventions, such as cognitive-behavioral therapy, have demonstrated effectiveness in treating individual symptoms of distress among youth who have experienced bullying, they tend to focus on clinical treatment after harm has occurred rather than on systemic prevention or early identification [24]. Importantly, school- and clinic-based models often overlook the broader ecosystem where youth experience bullying, leaving gaps in coordination between educational, health, and community systems [7,25].

Recognizing bullying and cyberbullying as public health issues highlights the need for integrated approaches that extend beyond schools and mental health settings [20,22,23]. Healthcare and public health professionals, such as physicians, school nurses, public health educators, and allied health providers, frequently interact with youth and are uniquely positioned to contribute to early identification, prevention, and respond to bullying-related harm [2,7,26]. In practice, these professionals may be the first to identify bullying-related distress during clinical visits or school-based health encounters. A study by Ranney et al. [27] found that pediatric clinicians regularly encounter bullying in practice but lack standardized definitions, screening tools, and referral protocols, underscoring the need for consistent clinical guidance and system-level coordination. Similarly, Celdrán-Navarro et al. [28] emphasized that bullying and cyberbullying research conducted within healthcare settings can more effectively address tertiary prevention and foster collaboration across healthcare, educational, and community systems. Framing bullying and cyberbullying as public health issues, therefore reflects the practical realities faced by clinicians and the growing recognition that prevention and intervention require coordinated, cross-sector approaches. Despite this, the role of healthcare and public health professionals in addressing bullying remains underexplored in the literature [7,29].

This systematic review addresses that gap by evaluating peer-reviewed empirical studies focused on interventions led by healthcare and public health professionals to address bullying and cyberbullying among youth. The review aims to (1) identify evidence-based approaches and strategies used in clinical and public health contexts to mitigate the effects of bullying and cyberbullying, (2) examine how resilience-building, trauma-informed care, and empowerment-based practices are integrated into these interventions, and (3) provide actionable recommendations for translating these findings into practical guidance for healthcare providers and public health professionals. To guide the analysis, the review is organized around three research questions:What evidence-based interventions and prevention strategies have healthcare and public health professionals employed to address bullying and cyberbullying among youth?How have healthcare and public health professionals implemented resilience-building, trauma-informed, and empowerment-based approaches within these interventions?What recommendations can be made for integrating these findings into clinical and public health practice?

## 2. Conceptual Framework

This review is guided by the SHIELD framework (Strengths, Healing, Interventions, Empowerment, Learning, Development), developed by Dailey and Roche [7] through a scoping review of 143 empirical studies. Grounded in established behavioral and ecological theories, including Social Learning Theory [30], Ecological Systems Theory [31], the Theory of Planned Behavior [32], and Moral Disengagement Theory [33], the framework integrates evidence on how individual cognitions, peer norms, and social environments interact to influence traditional and online aggression. SHIELD translates these theoretical principles into practice by emphasizing strengths, healing, and empowerment within health and public health contexts. It also draws from applied behavioral and social–emotional models such as the Collaborative for Academic, Social, and Emotional Learning (CASEL) framework and Positive Behavioral Interventions and Supports (PBIS) to guide implementation for healthcare and public health professionals [34,35]. By synthesizing findings across diverse interventions and theoretical approaches, the SHIELD framework provides a structured model for aligning prevention and intervention strategies with resilience-building and empowerment, serving as the guiding theory for organizing and interpreting findings in this review.

The “Strengths” domain highlights the role of health and mental health professionals in recognizing and amplifying youth’s existing assets. Through strength-oriented dialog and reflection, practitioners can help young people identify and apply their personal strengths to navigate challenges related to bullying. The “Healing” component emphasizes the importance of creating safe and supportive environments where youth can process their experiences of bullying and develop healthier coping mechanisms. The “Interventions” domain focuses on applying evidence-based strategies led by healthcare and public health professionals to prevent, identify, and respond to bullying and cyberbullying across various settings.

The framework also includes “Empowerment,” which focuses on building the confidence and skills of youth to advocate for themselves and others. This is supported through educational activities, role-playing, and peer or community-based engagement efforts. The “Learning” domain emphasizes the role of healthcare professionals in expanding youths’ understanding of bullying dynamics, promoting social-emotional learning, and enhancing coping skills. Lastly, the “Development” component addresses the need for sustained learning opportunities for youth, caregivers, and practitioners. This emphasis on ongoing learning fosters long-term, system-level responses to bullying and cyberbullying.

SHIELD is particularly relevant to public health practice because it addresses a critical gap in anti-bullying models by emphasizing the role of non-mental health professionals, such as physicians, nurses, school nurses, and health educators, as key players in prevention and intervention efforts [7]. While school-based anti-bullying frameworks are well-established, SHIELD provides a developmentally informed, cross-sector model that encourages collaboration among schools, families, and public health systems. The framework promotes holistic, culturally responsive care by connecting clinical expertise and community-based strategies with a focus on trauma-informed care, empowerment, and equity.

## 3. Methods

This systematic review was conducted following the Preferred Reporting Items for Systematic Reviews and Meta-Analyses (PRISMA) 2020 guidelines to ensure transparency and methodological rigor. The study protocol was registered on 8 September 2025 with the Open Science Framework (OSF). The full protocol is available at https://doi.org/10.17605/OSF.IO/5WKNB.

### 3.1. Eligibility Criteria

Studies were eligible for inclusion if they were peer-reviewed, published in English between 2013 and 2023, and focused on interventions for bullying or cyberbullying that involved healthcare or public health professionals, such as physicians, nurses, school nurses, public health professionals, health educators, or allied health providers, working with youth populations, including children, adolescents, or school-aged individuals. Eligible interventions included prevention programs, trauma-informed care, resilience training, health promotion, screening tools, and empowerment-based strategies relevant to healthcare or public health practice. Because contemporary prevention models increasingly address school bullying and cyberbullying together (e.g., integrated SEL, digital citizenship, and caregiver engagement), studies of either form were eligible when aligned with healthcare or public-health roles. Recent systematic and meta-analytic reviews demonstrate a growing convergence in intervention design and implementation across in-person and digital aggression [14,15,16].

Studies were excluded if they were non-empirical (e.g., literature reviews without original data, opinion pieces, or theoretical articles), focused exclusively on school-based interventions without a healthcare or public health component, or addressed mental health treatment in isolation without a broader public health or resilience-building emphasis. Additionally, studies that did not explicitly involve healthcare or public health professionals in implementing the intervention were excluded, along with single-case reports or studies lacking methodological rigor.

### 3.2. Information Sources and Search Strategy

A comprehensive literature search was conducted between February and March 2025 across three major databases: PubMed, Web of Science, and EBSCOhost (CINAHL and PsycINFO). Although Scopus was part of the initial search strategy, it was ultimately excluded due to access limitations. The search strategy combined controlled vocabulary (e.g., MeSH terms) and relevant keywords related to four core domains: bullying and cyberbullying, healthcare and public health professionals, intervention and prevention strategies, and youth populations. Boolean operators, truncation, and phrase searching were employed to optimize precision and coverage. The full database search strings are available in the OSF registration record (https://doi.org/10.17605/OSF.IO/5WKNB).

### 3.3. Study Selection Process

The study selection process adhered to PRISMA 2020 guidelines and involved three phases: title review, abstract review, and full-text review. A total of 561 records were retrieved from database searches, with an additional five studies identified through backward citation tracking, resulting in 566 records. After excluding 40 duplicates through automated and manual procedures, 526 unique articles remained.

In the first phase, two reviewers independently screened all 526 titles using predefined eligibility criteria. Articles were excluded if they did not address bullying or cyberbullying, did not involve healthcare or public health professionals, or did not focus on youth populations. This phase resulted in 168 articles progressing to abstract review. During the second phase, the abstracts were independently assessed for relevance to the review objectives, and decisions were documented through a collaborative process until consensus was reached. In the final phase, 42 full-text articles were evaluated for inclusion. At the full-text stage, 30 studies were excluded. Twelve did not involve healthcare or public health professionals working with children or youth, 10 were non-empirical, and 8 lacked an intervention focus. Twelve empirical studies were included in the final synthesis. The study selection process is summarized in Figure 1.

All 12 included studies underwent quality appraisal using the Joanna Briggs Institute (JBI) Critical Appraisal Tool, with studies evaluated based on their design (e.g., qualitative, quasi-experimental, scoping review). Two reviewers independently conducted the appraisals, with discrepancies resolved through discussion. Studies were scored out of 8 or 10, depending on the JBI version applied, and all met or exceeded a 75% quality threshold. In addition, risk of bias was assessed using adapted criteria appropriate for each study type. Table 1 summarizes the quality ratings, risk of bias ratings, study designs, and notable methodological considerations of each study.

### 3.4. Data Synthesis Approach

The data were synthesized using a narrative synthesis approach, which is well-suited for systematic reviews that include studies too diverse in design, population, interventions, and outcomes to be combined statistically [49]. Narrative synthesis systematically describes, compares, and interprets findings across studies using structured narrative rather than numerical aggregation. This approach is particularly useful for identifying patterns in both qualitative and quantitative studies and for understanding how and why complex, context-dependent interventions may be effective. In this review, themes were developed inductively through thematic analysis following Braun and Clarke [50]. This process included familiarization with the data, generating initial codes, searching for patterns, and developing and refining themes based on shared characteristics and outcomes across the interventions. The result was a set of seven core themes reflecting emerging practices and perspectives across the reviewed studies.

Once these themes were established, a post-synthesis mapping process was conducted using the SHIELD framework [7]. This mapping served as an interpretive lens to align the inductively developed themes with the six components of the SHIELD framework, a conceptual model designed to evaluate how health and mental health professionals support youth impacted by bullying and cyberbullying. Drawing on guidance regarding framework synthesis [51], this step added theoretical coherence to the findings by situating them within a structured public health model. The alignment process clarified how existing interventions correspond to SHIELD’s components and identified gaps in the literature, particularly in underrepresented areas such as empowerment and sustained development. These insights inform the review’s conclusions and highlight opportunities for enhancing the role of healthcare and public health professionals in bullying prevention and intervention efforts. A summary of the included studies, including study design, setting, professional roles, intervention focus, and key outcomes, is presented in Table 2.

## 4. Findings

Using a narrative synthesis approach [49] and thematic analysis [50], six key themes emerged from the 12 included studies. These themes highlight the diversity of interventions and strategies employed by healthcare and public health professionals to address bullying and cyberbullying among youth. The themes are (1) screening and early identification protocols, (2) family and community involvement, (3) variable focus on mental health and well-being, (4) multi-component, school-based interventions, (5) cognitive-behavioral and solution-focused interventions, and (6) online and digital interventions. These themes were intentionally organized to clearly distinguish between screening, prevention, and intervention efforts. For example, educational programs designed for groups where bullying has not occurred fall under prevention, while activities directed toward adolescents already involved in bullying or cyberbullying are classified as interventions. Below, these themes are presented with illustrative examples and synthesis across studies.

### 4.1. Theme 1: Screening and Early Identification Protocols

This theme pertains to the use of formal tools or professional observation by healthcare providers to identify youth who may be facing bullying or cyberbullying, thus enabling timely referral to appropriate support services and early intervention. Seven of the 12 reviewed studies provided clear or partial support for this theme [40,41,42,45,46,47,48]. Hutson et al. [41] emphasized the critical role of validated screening tools for identifying adolescents involved in cyberbullying yet noted a striking absence of such practices in healthcare settings, particularly among school nurses and primary care providers. Yoseph et al. [47] highlighted the importance of screening protocols for school-based healthcare staff to identify at-risk students early and facilitate timely referrals to counseling or support services, thereby reducing the likelihood of escalation and improving psychological outcomes. Hutson and Melnyk [40] further demonstrated the clinical utility of structured assessments in a pediatric behavioral health setting. Through regular monitoring by healthcare professionals of depression, anxiety, and bullying victimization, their intervention enabled early detection and individualized care for adolescents experiencing bullying-related distress.

Four additional studies provided partial support for this theme. Although they did not employ formal screening instruments, Hutson, Thompson, and Melnyk [42] utilized pre- and post-assessments to evaluate changes in bullying victimization and mental health symptoms, reinforcing the significance of outcome monitoring in early intervention. Yosep et al. [46,48] underscored the role of school health professionals (e.g., nurses, counselors) in recognizing early signs of bullying-related distress and initiating support, though specific screening tools or protocols were not described. Furthermore, while Öztürk, Çopur, and Kubilay [45] relied on teacher and nurse referrals to identify students for participation, demonstrating the value of cross-disciplinary collaboration in the absence of formal tools. Collectively, these studies indicate a growing recognition that early identification, whether through validated instruments or professional observation, is fundamental to effective and responsive bullying prevention in school and clinical settings.

### 4.2. Theme 2: Family and Community Involvement

This theme captures the involvement of families and community stakeholders in bullying prevention efforts through partnership and collaboration between healthcare staff who deliver the intervention and caregivers who reinforce skills at home, as well as broader interdisciplinary and community partnerships designed to promote sustainable outcomes. Eight of the 12 reviewed studies explicitly supported this theme, identifying family and community involvement as a key contributor to the effectiveness of bullying and cyberbullying interventions [41,42,45,46,47,48]. The remaining studies included partial support; for instance, Kvarme et al. [42] incorporated parent counseling, and Evgin and Bayat [38] emphasized teacher and peer support, aligning with broader principles of multi-contextual collaboration [41,47]. Several interventions featured structured caregiver involvement. In MINDSTRONG, for example, caregiver psychoeducation was included to reinforce skills learned by students in clinical sessions [40,42]. Similarly, Yosep et al. [46,48] identified caregiver engagement and community partnerships as central elements in effective cyberbullying interventions. Brandão Neto et al. [38] implemented a participatory model grounded in Freirean pedagogy that encouraged shared leadership among students, families, educators, and healthcare professionals.

Systematic reviews by Hutson et al. [41] and Yosep, Hikmat, and Mardhiyah [47] reinforced these findings, noting that the most effective interventions integrated multi-stakeholder collaboration, including parents and families, into the design and implementation of bullying prevention strategies. These studies align with broader ecological models that view bullying as a phenomenon influenced by interconnected systems of support, including family, school, and community. International evidence further supports these ecological levers. Research from China indicates that moral disengagement functions as a modifiable cognitive pathway to cyberbullying, suggesting that family- and community-focused components can enhance program impact [52]. Japanese meta-analytic findings highlight that coordinating schools and caregivers sustains social-emotional gains and improves program effectiveness [18]. Even among studies that did not prioritize family programming, elements of interdisciplinary collaboration (as emphasized in the fourth theme) and peer-adult support reflect foundational principles of a multi-contextual approach [39,43]. Such coordination enhances early identification, intervention, and continuity of care, enabling students to receive consistent messages and reinforcement across environments.

### 4.3. Theme 3: Variable Focus on Mental Health and Well-Being

This theme reflects the varying degrees to which bullying and cyberbullying interventions integrated mental health and emotional well-being, ranging from structured therapeutic approaches to more general social-emotional strategies. Across the 12 reviewed studies, most addressed psychological outcomes in some form, though the extent and focus of mental health integration varied. Several studies featured structured, evidence-based mental health components. Hutson and Melnyk [40] and Hutson, Thompson, and Melnyk [42] evaluated a cognitive behavioral therapy (CBT)-based intervention in a pediatric behavioral health setting that targeted emotional regulation and cognitive restructuring, incorporating caregiver involvement. Both studies reported reductions in depression, anxiety, and bullying victimization. Similarly, Öztürk Çopur and Kubilay [45] implemented nurse-led, solution-focused group therapy that improved students’ emotional self-efficacy and coping strategies.

Other studies emphasized emotional well-being through less formalized methods. Kvarme et al. [43,44] employed school nurses to implement group-based support and behavioral strategies to encourage emotional expression, reduce peer victimization, and foster social connectedness. Evgin and Bayat [39] used creative drama techniques implemented by school nurses to build empathy and communication skills in adolescents affected by bullying. Hutson et al. [41], in a systematic review of cyberbullying interventions, found that empathy training and social skills development implemented by healthcare providers (e.g., school nurses, primary care, mental health clinicians) were consistently linked to improved psychological outcomes. Yosep et al. [46,48] similarly noted self-confidence, emotional resilience, and symptom reduction were a key goal of school-based and nurse-led interventions, while Avşar and Alkaya [37] addressed internalizing symptoms through assertiveness training led by nurses. Yosep, Hikmat, and Mardhiyah [48] further emphasized the role of school health professionals in addressing trauma and called for stronger integration of mental health into school prevention frameworks. Across these studies, the focus on mental health ranged from central clinical objectives to broader social-emotional outcomes, illustrating diverse approaches to addressing psychological well-being within bullying prevention efforts.

### 4.4. Theme 4: Multi-Component, School-Based Interventions

This theme refers to interventions delivered collaboratively by multiple stakeholders (e.g., educators, school nurses, counselors, parents, and students) in school settings. These programs combine psychoeducation, skill-building, and mental health promotion to prevent and respond to bullying and cyberbullying. Program duration ranged from single-session workshops to multi-week group interventions [37,38,39,40,43,44,45], although most studies did not specify standardized implementation periods or follow-up timelines [40,45,46,47,48]. Nine of the 12 reviewed articles were particularly effective in addressing bullying and peer victimization [38,39,42,43,44,46,47,48], especially when they incorporated psychoeducation, social-emotional skill development, peer leadership, and the active engagement of health professionals. While the remaining four studies provided partial support for multi-component, school-based approaches, they still reflected elements relevant to this theme, such as mental health programming, school nurse involvement, or behavioral strategies.

In a systematic review, Yosep et al. [47] underscored that the most impactful programs extend beyond traditional school-centered models by incorporating family, health, and community partnerships to support long-term sustainability. Large-scale trials such as the SEHER intervention in India [53] and the INCLUSIVE program in the UK [54] further demonstrated that systems-level, multi-stakeholder strategies can improve school climate and promote student well-being. Building on this systems-level perspective, Kvarme et al. [43] emphasized the importance of collaboration between students, teachers, and school health professionals to foster empathy and social support. Similarly, Hutson, Thompson, and Melnyk [42] advocated for integrated models combining health education, resilience-building, and parental involvement, positioning school nurses and health professionals as key agents in prevention and response.

The Planning the Antibullying Program of Health Education (PATES) exemplified this integrative model, combining youth-led leadership and co-created interventions among students, educators, and health professionals [38], reinforcing the value of inclusive, interdisciplinary collaboration [38,48]. Several studies in this review highlighted the importance of specific intervention components, such as empathy training, digital citizenship, and coping strategies, in reducing cyberbullying and promoting well-being. Hutson et al. [41] and the National Academies of Sciences, Engineering, and Medicine [26] emphasized caregiver education as a critical feature for expanding the reach and sustainability of such efforts. Yosep et al. [46] similarly stressed the roles of healthcare workers and community partners in implementing empathy-based interventions and delivering school counseling services.

Interactive and experiential learning methods, such as creative drama, role-playing, and collaborative activities, were frequently used to promote empathy, problem-solving, and peer engagement. For example, Yosep et al. [48] described innovative practices like ACT OUT! Social Issue Theater and game-based interventions that position nurses as both facilitators and evaluators. Results indicated these approaches enhanced the school climate and strengthened students’ relational support systems [39,45]. Youth-centered, dialogic approaches like Paulo Freire’s Culture Circles reinforce the student agency’s role in prevention efforts [38]. These participatory models foster shared responsibility, prosocial behavior, and collective action. Avsar and Alkaya [37] further highlighted the value of peer-led assertiveness training and communication skill-building, especially in role-play-based interventions.

### 4.5. Theme 5: Cognitive-Behavioral and Solution-Focused Interventions

This theme encompasses interventions grounded in CBT and solution-focused approaches that aim to reduce bullying-related distress by helping youth build coping strategies, regulate emotions, and develop problem-solving and interpersonal skills. Five of the 12 reviewed articles featured CBT and solution-focused interventions as effective strategies for addressing bullying and its associated psychological consequences [40,42,43,44,45]. CBT-based programs, such as MINDSTRONG (adapted from COPE), combined psychoeducation, structured group delivery, and caregiver engagement to reduce symptoms of anxiety, depression, and peer victimization [40,42], and were typically facilitated by school nurses or mental health professionals within school or clinical settings. Similarly, solution-focused brief interventions emphasized goal-directed, strengths-based problem-solving and were associated with enhanced resilience and decreased victimization when delivered by trained health professionals [43,45]. These findings are bolstered by broader evidence syntheses, including the systematic review by Hutson et al. [41], which identified CBT and solution-focused methods as essential components of effective bullying and cyberbullying interventions, and the scoping review by Yosep et al. [46], which emphasized empathy-based counseling and the role of health professionals in delivering interventions that promote psychological well-being and reduce bullying behaviors. Across these studies, cognitive-behavioral and solution-focused models provided structured, evidence-based approaches to addressing both the emotional and behavioral dimensions of bullying.

### 4.6. Theme 6: Online and Digital Interventions

This theme refers to bullying prevention and support strategies delivered through digital platforms, such as mobile apps, web-based modules, or virtual counseling, with a focus on accessibility, scalability, and engagement. These interventions were typically facilitated by healthcare or public health professionals in clinical, school-linked, or community settings. Six of the 12 reviewed studies examined online or digitally delivered interventions targeting youth affected by bullying or cyberbullying [40,41,42,46,47,48]. These interventions addressed domains such as emotional regulation, coping skills, social-emotional learning, and resilience-building.

Several programs demonstrated positive outcomes in clinical settings. For example, Hutson and Melnyk [40] evaluated MINDSTRONG to Combat Bullying; an online CBT-based intervention delivered in pediatric behavioral health contexts. The program included psychoeducation, structured skills training, and caregiver engagement and was associated with reduced depression, anxiety, and perceived victimization. Other studies focused on synthesizing digital intervention characteristics. For example, Hutson et al. [41] and Yosep et al. [46] identified digital features, such as digital citizenship training, interactive skill-building modules, and caregiver-focused content, as key to program effectiveness. Yosep et al. [47] further documented the use of asynchronous counseling, peer interaction, and multimedia tools, often used in collaboration with schools or community organizations. Although not part of the reviewed studies, evidence from a South Korean quasi-experimental trial similarly demonstrated that a school-delivered online cyberbullying education program reduced both perpetration and victimization while increasing defender behavior among elementary students [55]. Despite these promising findings, across all three reviews [41,46,47], a consistent implementation gap was identified, with health-led interventions often operating independently and integration within broader healthcare or public-health systems remaining limited.

The remaining seven studies also included components like group-based counseling, skills training, and behavior management that align closely with elements found in digital formats. Several digital programs, including CORE and Cyberprogram 2.0, were initially developed for in-person delivery and later adapted for virtual platforms without compromising effectiveness [47].

## 5. Discussion

The first research question sought to identify evidence-based interventions and prevention strategies healthcare and public health professionals employ to address bullying and cyberbullying among youth. The findings reveal that while interventions vary in format and delivery setting, they share core features grounded in psychological theory, public health principles, and interdisciplinary coordination [40,45,47]. These features align with the SHIELD framework’s Interventions, Healing, and Empowerment components, which emphasize integrated, trauma-informed, and strength-based responses to youth distress [7]. In several studies, healthcare professionals, particularly school-based health and mental health professionals, played pivotal roles in delivering or coordinating comprehensive, multi-component interventions [44,48]. These programs combined psychoeducation, social-emotional learning, peer engagement, and caregiver involvement which were associated with improvements to student well-being and school climate [40,47,48]. Whether facilitating group sessions, monitoring progress, or making referrals, health professionals often act as the connective tissue across home, school, and clinical environments [39,45]. School-based and interdisciplinary programs were reported to have higher feasibility and sustainability, especially when implemented through existing health service infrastructure [44,46].

A central feature of many interventions was the incorporation of structured, evidence-based therapeutic frameworks. Cognitive-behavioral and solution-focused approaches were often employed to reduce internalizing symptoms, strengthen emotional regulation, and build resilience among youth exposed to bullying [40,43,45]. These approaches align with SHIELD’s Healing and Development components by fostering self-efficacy and emotional well-being through structured, goal-directed support [7]. Typically delivered by nurses and behavioral health professionals, these models offered potentially feasible options for addressing the psychological sequelae of victimization while offering feasible options for implementation in pediatric, school-linked, or community health settings [40,45]. Although designed for individual-level change, such interventions were often embedded within broader systems that emphasized coordination with school personnel and families [40,43].

Digital and hybrid models emerged as a promising yet underutilized approach [41,46]. A subset of studies reviewed online interventions designed to enhance access and reduce barriers for youth affected by cyberbullying [41,46]. These digital strategies often reflected SHIELD’s Learning and Development dimensions by expanding students’ access to knowledge, coping strategies, and support beyond the traditional school setting [7]. These tools were frequently derived from in-person programs originally developed by health professionals and adapted for broader, asynchronous reach [47]. Although implementation within healthcare systems remains limited, the scalability and flexibility of digital modalities were emphasized as particularly relevant for dispersed or underserved populations [40,41,42,46,47,48].

Although fewer included studies focused exclusively on cyberbullying, this review intentionally retained these studies, along with those addressing both bullying and cyberbullying, because healthcare and public-health interventions increasingly target both contexts within the same frameworks. Evidence from recent systematic and meta-analytic reviews [14,15,16] suggests convergence in program components such as social-emotional learning, empathy training, digital citizenship, and caregiver engagement, as well as in delivery across settings. Retaining these studies is therefore consistent with the review’s a priori scope (health- and public-health–led interventions) and aligns with current multidisciplinary practice.

Another important element across interventions was the role of health professionals in identifying and responding to early indicators of bullying-related distress [40,41]. While only a few studies incorporated formalized screening tools, several described pre/post assessments, teacher-nurse collaboration, or informal referral mechanisms that enabled timely intervention [39,40]. These practices are aligned with SHIELD’s Strengths and Interventions components, as they emphasize proactive recognition of risk and the mobilization of supports that build resilience and reduce harm [7]. In school settings especially, the familiarity and visibility of nurses enabled early recognition of psychological symptoms, even when validated instruments were not consistently applied [47].

Family and community engagement is often recognized as crucial to effective intervention delivery [41,45]. Professionals work alongside caregivers through structured outreach, skill-building sessions, and culturally sensitive communication strategies [41,48]. These practices illustrate the SHIELD framework’s focus on Empowerment by equipping families with the tools and knowledge to support their children, and Development by creating strong, supportive ecosystems around youth [7]. Interventions that involved caregivers were generally associated with stronger engagement, more consistent follow-through, and an increased capacity for long-term behavioral change at home and in school [41,48]. This focus on multi-contextual coordination reflects a growing consensus that bullying prevention must address not only the individual but also the broader ecology of support surrounding the child. Together, these findings highlight how health professionals function as key connectors across systems, linking individual care, family engagement, and school-based supports, and align within SHIELD framework in the promotion of resilience, healing, and collaboration [7].

The second research question shifted its focus to understanding what applied practices for implementation can be derived from current literature to enhance the role of healthcare and public professionals in bullying prevention. Synthesizing the findings from various studies reveals not only promising intervention components, but also systemic gaps, underutilized opportunities, and critical considerations for improving implementation, equity, and sustainability across care settings [41,46,47,56].

Despite promising outcomes, several systemic barriers limited implementation and scalability across contexts. Multiple studies highlighted insufficient training and workload constraints among nurses and allied health professionals, which hindered sustained delivery of prevention programs [46,47]. Resource limitations, particularly lack of institutional support, funding, and time allocation, were also noted as key barriers to integrating interventions into routine health practice [39,45]. Fragmented coordination between health and education systems further reduced continuity of care and interprofessional collaboration, as interventions often operated independently rather than through formal partnerships [40,43]. These challenges mirror broader implementation issues identified in global public health literature, underscoring the need for structured funding mechanisms, protected training time, and cross-sector integration to sustain evidence-based bullying prevention efforts within healthcare infrastructures [14,15,54,55].

Findings from this review suggest several best practices that can guide healthcare and public health professionals in strengthening their role in bullying prevention [40,43,47]. Foremost, integrating mental health support as a foundational element of intervention design. This directly reflects SHIELD’s focus on Healing and Intervention, underscoring the importance of addressing trauma and psychological distress in youth populations [7]. Given the consistent psychological consequences of bullying, health-led efforts must center on developmentally appropriate, trauma-informed approaches [41,45,46]. Programs like MINDSTRONG to Combat Bullying demonstrate how CBT-based frameworks can effectively reduce internalizing symptoms when implemented in pediatric and school-linked settings [40,48]. At the same time, the review revealed variability in the depth and consistency of mental health integration, highlighting the need for more systematic application of therapeutic principles across contexts [46,47].

Building on this foundation, the accessibility of school-based health professionals, including nurses and behavioral health providers, should be leveraged more fully. School nurses and behavioral health providers frequently serve as the first points of contact for students experiencing bullying [45,47]. SHIELD’s Strengths and Empowerment dimensions emphasizes the importance of mobilizing trusted adults and professional networks to build youth capacity and resilience [7]. To maximize their impact, training school nurses and behavior health providers in bullying-related screening, solution-focused techniques, and trauma-responsive care should be expanded.

Interventions were most effective when they were multi-component and encouraged collaboration among health providers, educators, caregivers, and community leaders [41,45], aligning closely with the SHIELD components of Development and Empowerment, promoting relational and systemic supports that extend beyond individual-focused strategies [7]. Programs that integrated psychoeducation, social-emotional learning, family communication, and community support tendered to report more favorable outcomes [47,48]. However, efforts to identify and support youth early in the intervention process were inconsistent across studies. Although some programs used pre/post assessments or informal referrals, few employed validated screening tools to detect bullying involvement or track intervention outcomes [40,41]. Embedding routine screening into pediatric visits and school health assessments could facilitate earlier detection and personalized care, an approach that aligns with SHIELD’s Intervention and Strengths principles by supporting proactive, context-sensitive responses [7].

Digital platforms also emerged as a promising avenue for extending the reach of evidence-based interventions. Web-based CBT, interactive skill-building modules, and mobile counseling tools show promise for accessibility and scalability, particularly for youth in rural or underserved communities [41,46,47]. These innovations reflect SHIELD’s Learning and Development elements by increasing access to social-emotional knowledge, digital literacy, and mental health resources in youth-friendly formats [7]. Despite this promise, few of these interventions have been formally integrated into public health or clinical systems, underscoring the need for ethical, secure, and effective digital delivery models.

Finally, our third research question aimed to leverage our findings and identify recommendations for integrating these themes into medical and public health practice. While cognitive-behavioral and solution-focused interventions (Theme 5) have demonstrated positive outcomes, their effectiveness is maximized when adapted to reflect the lived experiences, cultural backgrounds, and identity-related stressors encountered by diverse youth populations [45]. For example, Hutson and Melnyk [42] adapted the MINDSTRONG program by incorporating psychoeducation, culturally sensitive examples, and caregiver engagement components, resulting in significant reductions in depressive symptoms, anxiety, and perceived victimization. Similarly, Yosep et al. [46] modified psychoeducational nursing interventions to align with collectivist family norms and communication patterns, which improved participant engagement and intervention sustainability. These culturally responsive adaptations reflect SHIELD’s Healing and Empowerment components, emphasizing trauma-informed care that affirms identity and fosters agency in youth [7].

Similarly, school-based, multi-component programs (Theme 4) are more effective when they actively involve families and communities in ways that honor linguistic, cultural, and social differences, ensuring that engagement is not only present but also empowering [48]. Brandão Neto et al. [38] employed a Freirean participatory pedagogy in Brazil that emphasized shared leadership among students, families, and healthcare professionals to reflect community values of dialogue and empowerment. In Indonesia, Yosep et al. [46] adapted psychoeducational nursing interventions to align with local communication norms and collective family decision-making. Similarly, Evgin and Bayat [39] incorporated empathy-building and peer collaboration within a framework aligned with Turkish cultural expectations of teacher–student relationships, while Kvarme et al. [43] highlighted the importance of inclusive, relational approaches rooted in Norwegian cultural norms. These studies demonstrate that culturally grounded improves participation and reinforces the ecological and empowerment principles central to the SHIELD framework [7].

Digital interventions (Theme 6) provide scalable and flexible solutions, particularly for reaching youth in rural, marginalized, or hard-to-reach communities. When implemented with intention, digital models can expand access to high-quality, culturally responsive tools that support resilience and emotional growth [7,46]. However, their content and delivery must be intentionally designed to avoid reproducing inequities and to reflect inclusive, developmentally appropriate messaging. Studies have identified ongoing challenges such as inconsistent internet connectivity, limited digital literacy, and restricted infrastructure in lower-resource settings, all of which reduce equitable participation [47,55]. Broader reviews of digital health interventions confirm that these barriers persist due to underinvestment in technology, insufficient staff training, and ethical uncertainties regarding confidentiality and data security [41,57]. Regional analyses further demonstrate that unequal access and resource distribution across schools and communities constrain effectiveness, especially in Asia-Pacific contexts [58]. At the same time, success examples such as Choi et al. [56], who reported significant reductions in cyberbullying behaviors following an online school-based program in South Korea, and Hutson and Melnyk [39], whose web-based CBT model improved accessibility in pediatric behavioral health, underscore the feasibility of ethically and culturally tailored digital approaches when designed for context and equity. These findings highlight the importance of coordinated, secure, and well-resourced digital delivery systems to ensure that interventions advance rather than widen health and educational disparities.

The importance of screening and early identification (Theme 1) must also be understood through a culturally informed lens. Tools and referral protocols should be sensitive to systemic bias and adapted for contextual validity across diverse populations [41,47]. This reflects SHIELD’s Strengths and Intervention components, emphasizing early, individualized support rooted in recognizing youth assets and vulnerabilities [7]. Even family engagement strategies (Theme 2) benefit from equity-centered design, ensuring that caregivers from all backgrounds are equipped to support their children and are treated as essential partners rather than passive recipients of information. These strategies align with SHIELD’s focus on Empowerment and Development, reinforcing the value of shared leadership in prevention and care [7]. Ultimately, the theme of mental health and emotional well-being (Theme 3) cannot be fully realized without addressing the structural and identity-based factors that shape youth experiences of both bullying and access to care. The SHIELD framework calls for integrated models that are evidence-based, equity-driven, youth-informed, and systems-oriented, positioning healthcare professionals to act as interventionists and advocates for systemic change [7].

### Limitations and Recommendations for Practice and Research

Although the reviewed interventions show promise for addressing bullying and cyberbullying, several limitations should be considered when interpreting the results. Across studies, methodological and contextual constraints limited the generalizability of findings. Many employed small samples or non-randomized designs, as noted by Hutson and Melnyk [42] and Kvarme et al. [43], reducing internal validity and preventing robust causal inferences. Others, such as Evgin and Bayat [39], lacked control groups or used single-branch classroom settings, restricting the scope for comparison. In several cases, program participation was limited to a single site or district [37,38].

Several studies also reported challenges with implementation fidelity, participant diversity, or measurement rigor. For example, the empathy scale used by Evgin and Bayat [39] demonstrated low reliability, while Kvarme et al. [43] noted potential social desirability bias in self-reported bullying data. Scoping and systematic reviews by Yosep et al. [47,48] further highlighted constraints imposed by narrow inclusion criteria (e.g., limiting to the past ten years or nursing-only interventions), as well as the lack of integration across health, education, and community sectors. These patterns suggest that despite positive findings, many interventions remain context specific.

The review’s inclusion criteria also narrow interpretation. Only studies explicitly led by healthcare or public health professionals were included, which aligns with the research question but may have excluded relevant multidisciplinary or education-based programs. The focus on English-language, peer-reviewed publications may have missed work published in regional or non-English outlets. The absence of studies from East Asia likely reflects these disciplinary and language limitations rather than a lack of research activity in the region.

Overall, these limitations underscore that the evidence base for health-led bullying interventions remains emergent rather than definitive. Future studies should apply rigorous designs, such as randomized controlled trials, mixed-methods approaches, and multi-site replications, with larger samples, consistent outcome measures, and longer follow-up periods. Broader database searches and inclusion of gray literature would strengthen comparability, address potential bias, and provide a more comprehensive understanding of effective, sustainable health- and public health–led bullying prevention programs.

## 6. Conclusions

This systematic review highlights the expanding yet underutilized role of healthcare and public health professionals in bullying and cyberbullying prevention. While school-based interventions remain foundational, the evidence demonstrates that health professionals are uniquely positioned to deliver trauma-informed, resilience-building, and empowerment-based strategies beyond traditional educational settings. The reviewed studies illustrate how multi-component, collaborative approaches, particularly those grounded in cognitive-behavioral frameworks, digital platforms, and family engagement, can support youth across home, school, and clinical environments. Using the SHIELD framework [7] as an interpretive lens, this review underscores the importance of integrated, cross-sector models centered on healing, development, and empowerment in bullying intervention efforts. As the field moves forward, enhancing implementation systems to ensure fidelity, more equitable access to services, and greater inclusion of health professionals in prevention policy and practice are essential for creating safer, more supportive environments for all youth.

## Figures and Tables

**Figure 1 ijerph-22-01682-f001:**
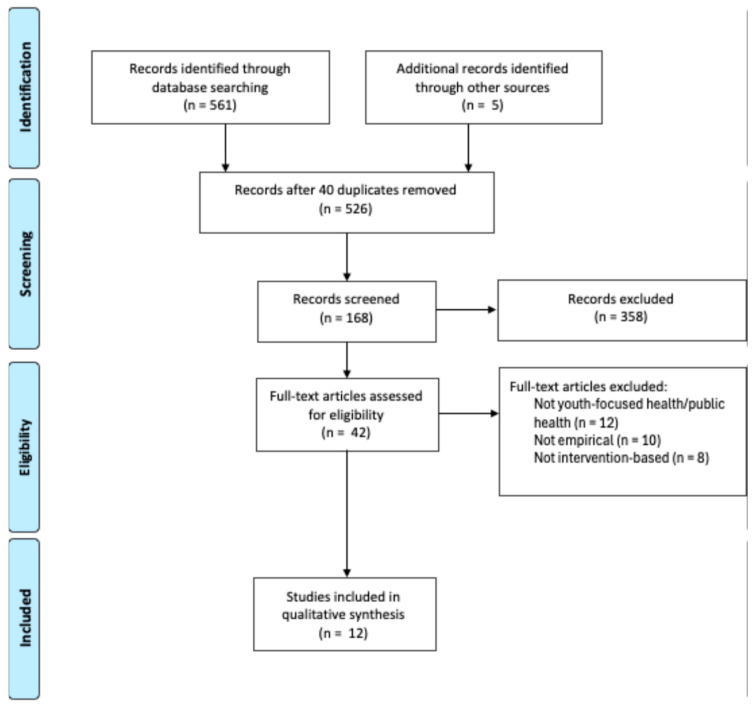
PRISMA Table [36].

**Table 1 ijerph-22-01682-t001:** Study quality and risk of bias (N = 12).

Author(s) and Year	Design	Quality Rating	R/B Rating	Considerations
Avşar & Alkaya [37]	Quasi-experimental	7/8	Some concerns	Assertiveness and peer support focus; limited design controls
Brandão Neto et al. [38]	Qualitative	10/10	Low	Semi-structured interviews with strong alignment to research aims
Evgin & Bayat [39]	Quasi-experimental	8/8	Low	Standardized tools and full protocol adherence
Hutson & Melnyk [40]	One-group pre/post	6/8	Some concerns	No control group; validated scales used
Hutson et al. [41]	Systematic review	3/3	High	Synthesized cyberbullying best practices
Hutson, Thompson & Melnyk [42]	Feasibility study	6/8	High	No comparison group; notable pre-post improvements
Kvarme et al. [43]	Quasi-experimental	6/8	Some concerns	No randomization; valid measurement tools
Kvarme et al. [44]	Qualitative	10/10	Low	Rich participant representation; thematic depth
Öztürk Çopur & Kubilay [45]	Quasi-experimental	7/8	Some concerns	Validated instrument used; no randomization
Yosep et al. [46]	Scoping review	3/3	Low	Well-defined categories of nursing intervention types
Yosep et al. [47]	Scoping review	3/3	Low	Transparent synthesis methods for nursing interventions
Yosep, Hikmat & Mardhiyah [48]	Scoping review	3/3	High	School-based cyberbullying program review

Note. Quality Rating reflects the number of “Yes” responses on the Joanna Briggs Institute (JBI) Critical Appraisal Checklist for the relevant study design (e.g., 7/8 indicates seven criteria met out of eight total). Risk of Bias (R/B) Rating reflects overall assessment of potential bias within each study, considering design limitations, sampling methods, and analytic rigor. Ratings are categorized as Low, Some concerns, or High risk of bias, consistent with common systematic review practices.

**Table 2 ijerph-22-01682-t002:** Study characteristics.

Study	Year	Sample (*n*)	Location	Bullying Type	Study Design	Professional Focus	Intervention Type
Avşar & Alkaya [37]	2017	Adolescents 12–14 (*n* = 64)	Turkey (Aydin)	Bullying	Pre/Post	School Health	Assertiveness Training
Brandão Neto et al. [38]	2020	Adolescents (*n* = 32)	Brazil (Pernambuco)	Bullying	Participatory Action Research	Public Health	Participatory Youth Leadership Program
Evgin & Bayat [39]	2020	Adolescents (*n* = 50)	Turkey (Izmir)	Bullying	Quasi- Experimental	School Health	Drama-Based Peer Learning
Hutson & Melnyk [40]	2022	Adolescents 13–17 (*n* = 52)	USA (Midwest)	Both	RCT	Nursing Behavioral Health	CBT-based Mental Health Program
Hutson et al. [41]	2018	Children & Adolescents (25 articles)	USA (Nationwide)	Cyber-bullying	Systematic Review	General Healthcare Providers	Systematic Review of Cyberbullying Interventions
Hutson, Thompson & Melnyk [42]	2022	Adolescents 13–17 (*n* = 63)	USA (Midwest)	Both	Quasi- Experimental	Nursing Behavioral Health	CBT-based Mental Health Program
Kvarme et al. [43]	2016	Children 11–12 (*n* = 16)	Norway (Trondheim region)	Bullying	Pre/Post	School Health	Group Counseling for Bullying Victims
Kvarme et al. [44]	2020	Adolescents 13–14 (*n* = 40)	Turkey (Western Anatolia)	Bullying	Quasi- Experimental	School Health	Creative Drama and Behavioral Education
Öztürk Çopur & Kubilay [45]	2022	Adolescents 11–14 (*n* = 14)	Turkey (Afyonkarahisar)	Bullying	Pre/Post	Nursing	Solution-Focused Group Counseling
Yoseph et al. [46]	2022	Children & Adolescents (17 articles)	Indonesia (Countrywide)	Both	Systematic Review	School Health	Systematic Review of School-Based Interventions
Yosep et al. [47]	2023	Children & Adolescents (18 articles)	Indonesia (Countrywide)	Bullying	Scoping Review	Nursing	Scoping Review of School Nurse Role
Yoseph, Hikmat & Mardhiyah [48]	2023	Children & Adolescents (11 articles)	Indonesia (Countrywide)	Bullying	Narrative Review	Nursing School Health	Narrative Review on Mental Health Integration

Note. RCT = Randomized Controlled Trial. For review studies, sample size reflects the number of articles included. Bullying Type indicates whether the intervention targeted traditional bullying, cyberbullying, or both.

## Data Availability

Extracted data, risk of bias ratings, and analytic notes are openly available in the OSF registration record (https://doi.org/10.17605/OSF.IO/5WKNB).

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
