# Peer review of "Addressing Bullying and Cyberbullying in Public Health: A Systematic Review of Interventions for Healthcare and Public Health Professionals"

_ijerph, 2025, doi:10.3390/ijerph22111682_

Round 1

Reviewer 1 Report

Comments and Suggestions for Authors
  1. Among the 12 included studies, most are studies about school bullying, while studies on cyber-bullying are relatively fewer. There are differences between cyber-bullying and school bullying in terms of causes and interventions. It is recommended that the current manuscript should focus on the field of school bullying.
  2. This review is guided by the SHIELD framework (Strengths, Healing, Interventions, Empowerment, Learning, Development), developed by Dailey and Roche. The order of the six components can be adjusted. It is suggested that the findings section could be written in the order of (1)Screening and Early Identification Protocols, (2)Prevention, and (3)Interventions.
  3. The author should pay attention to the distinction between prevention and interventions, as they target different groups. Education for groups where bullying has not occurred should be considered prevention, while educational activities for adolescents involved in bullying should be considered interventions.
  4. The finding parts need to be elaborated in more detail. Intervention programs should include who the implementers are, how different implementers (such as school teachers, healthcare and public health professionals) collaborate, the duration of the programs, the intervention methods, and the intervention effects.
  5. It may be worthwhile to include research findings from non-English-speaking countries, such as those from China orJapan, where many relevant studies have been conducted. It is hoped that the researchers will consider incorporating these findings.

Author Response

We are grateful for the reviewers’ thorough and constructive feedback. The manuscript has been substantially revised to improve clarity, balance, and critical analysis. 

Key revisions include:

  • Removal of unnecessary subheadings in the Introduction.
  • Integration of multiple theoretical frameworks within the Conceptual Framework section.
  • Condensation of redundant narrative elements.
  • More explicit cross-comparisons of intervention effectiveness.
  • Strengthened discussion of systemic and equity-focused implications.
  • Expanded and contextualized Limitations section.

A list of Reviewer 1 comments and actions taken to address each comment are provided below. Thank you again for this opportunity to improve our submission. 

Reviewer 1 Comments

Author Response

Among the 12 included studies, most are studies about school bullying, while studies on cyber-bullying are relatively fewer. There are differences between cyber-bullying and school bullying in terms of causes and interventions. It is recommended that the current manuscript should focus on the field of school bullying.

Thank you for this helpful comment. We acknowledge that fewer included studies focused exclusively on cyberbullying; however, we intentionally retained Hutson et al., 2018 and mixed bullying/cyberbullying studies (Hutson, Thompson & Melnyk, 2022; Hutson & Melnyk, 2022; Yosep et al., 2022) because healthcare and public-health interventions increasingly target both contexts within the same frameworks. Evidence from recent systematic and meta-analytic reviews (Fraguas et al., 2021; Gaffney et al., 2019; Ng et al., 2020) shows convergence in program components (e.g., SEL, empathy training, digital citizenship, caregiver engagement) and delivery across settings. Therefore, keeping these studies was an intentional decision consistent with our a priori scope (health- and public-health interventions) and with emergent multidisciplinary practice.

To address that our original text did not clearly articulate the inclusion of both bullying and cyberbullying within the review’s scope, we have now clarified this distinction and rationale in three sections: (1) Introduction §1.1 (added sentence on unified frameworks), (3) Methods §4.1 Eligibility Criteria (added rationale for including both forms when aligned with healthcare/public-health roles), and (4) Discussion §6 (new paragraph reiterating the convergence rationale with citations).

This review is guided by the SHIELD framework (Strengths, Healing, Interventions, Empowerment, Learning, Development), developed by Dailey and Roche. The order of the six components can be adjusted. It is suggested that the findings section could be written in the order of (1) Screening and Early Identification Protocols, (2) Prevention, and (3) Interventions.

Thank you for this thoughtful suggestion. In response, we reorganized the Findings section to improve conceptual flow and alignment with the SHIELD framework’s developmental sequence. The revised order now begins with: Screening and Early Identification Protocols, followed by Family and Community Involvement, Variable Focus on Mental Health and Well-Being, Multi-Component, School-Based Interventions, Cognitive-Behavioral and Solution-Focused Interventions, and Online and Digital Interventions. This revised sequence emphasizes early detection, prevention through family and community systems, and progressively complex intervention strategies, thereby enhancing clarity and theoretical coherence.

The author should pay attention to the distinction between prevention and interventions, as they target different groups. Education for groups where bullying has not occurred should be considered prevention, while educational activities for adolescents involved in bullying should be considered interventions.

Thank you for this thoughtful comment. We revised the Findings section to better distinguish between prevention and intervention efforts by clarifying definitions and reordering themes to reflect a developmental progression from screening and prevention to intervention. An explanatory sentence was also added to explicitly define prevention as programs for groups not yet affected by bullying and intervention as activities directed toward youth already involved. We hope these changes align with the reviewer’s recommendation and enhance conceptual and structural clarity throughout the section.

The finding parts need to be elaborated in more detail. Intervention programs should include who the implementers are, how different implementers (such as school teachers, healthcare and public health professionals) collaborate, the duration of the programs, the intervention methods, and the intervention effects.

We appreciate this suggestion and have revised the manuscript to clarify, where appropriate, who implemented each intervention (e.g., school nurses, public health educators, behavioral health professionals), how collaboration occurred among healthcare, education, and community sectors, and to provide additional detail on program duration, delivery methods, and intervention outcomes. Table 2 was also updated to summarize these features, offering a clearer picture of how different implementers and methods contributed to program effectiveness.

It may be worthwhile to include research findings from non-English-speaking countries, such as those from China or Japan, where many relevant studies have been conducted. It is hoped that the researchers will consider incorporating these findings.

Thank you for this helpful comment. We agree that including research from non-English-speaking contexts enhances the manuscript’s global relevance. We have now incorporated peer-reviewed East Asian findings that align with integrated intervention frameworks. Specifically, the Introduction (§1.1) references Liu et al. (2023), which identified moral disengagement and empathy as mechanisms in Chinese adolescents’ cyberbullying, and Takizawa et al. (2023), a Japanese meta-analysis highlighting school–family coordination in SEL implementation. To contextualize these additions, we also expanded Section §5 (Online and Digital Interventions; Family and Community Involvement) to connect international evidence for online education and ecological collaboration models, and Section §6.1 (Limitations) to clarify that the absence of East Asian studies among included articles likely reflects disciplinary orientation rather than a lack of regional research.

Reviewer 2 Comments

Author Response

The introduction needs to be revisited. There's no need for subheadings within the introduction. The introduction can end with research questions without including a "Rationale and Purpose" subheading. Avoid repetition as much as possible.

Thank you for this feedback. We have revised the introduction to remove unnecessary subheadings, integrate the “Rationale and Purpose” content into the main narrative, and streamline the text to reduce repetition. The introduction now concludes directly with the research questions for improved flow and coherence.

The reason for including healthcare and public health professionals in the research objective should be better explained with references.

Thank you for this helpful comment. We expanded the Introduction to better explain the rationale for focusing on healthcare and public health professionals, supported by new references. The revised text clarifies that clinicians and allied health providers often encounter youth experiencing bullying-related distress in clinical and school-based settings and are well positioned to support early identification, prevention, and intervention. This rationale is now substantiated with evidence from Ranney et al. (2016) and Hutson et al. (2018), which emphasize the need for coordinated, health-led approaches to bullying prevention and response.

Conceptual Framework: It focuses on SHIELD, whereas the topic is bullying and cyberbullying. It is necessary to provide a theoretical background on this issue.

Thank you for this insightful comment. We have revised the Conceptual Framework to explicitly include the theoretical foundations that underpin SHIELD. The section now describes how the framework integrates Social Learning Theory, Ecological Systems Theory, the Theory of Planned Behavior, and Moral Disengagement Theory, along with applied models such as CASEL and PBIS. This addition clarifies that SHIELD extends established theories of bullying and behavior change to healthcare and public health contexts, ensuring theoretical alignment with the topic of bullying and cyberbullying.

Discussion: Streamline the narrative: Reduce redundancy by consolidating overlapping discussions of trauma-informed care, digital interventions, and family engagement into tighter thematic syntheses. Certain themes (e.g., trauma-informed care, interdisciplinary collaboration, digital interventions) are repeated across multiple sections, leading to redundancy. Condensing would improve readability.

We appreciate this suggestion and have revised the manuscript to consolidate overlapping discussions of trauma-informed care, digital interventions, and family engagement for greater coherence. Redundant material was reduced, and these concepts are now presented within tighter thematic syntheses to enhance readability and narrative flow.

Discussion: Strengthen critical appraisal: The review sometimes overstates the effectiveness of interventions without sufficiently qualifying the strength of evidence (e.g., “consistently demonstrated stronger outcomes” may not be justified if based on small, non-randomized samples).

Thank you for this valuable feedback. We carefully reviewed Sections 5 (Findings) and 6 (Discussion) to ensure all claims about intervention effectiveness are appropriately qualified. Terms such as “strong” and “consistent” were replaced with “promising,” “positive,” or “context-specific.” We also expanded Section 6.1 (Limitations) to discuss methodological weaknesses (e.g., small samples, non-randomized designs, and limited generalizability), ensuring that interpretation of findings accurately reflects the strength of the underlying evidence.

Discussion: Provide more explicit comparisons of effectiveness across intervention types. For example: Which interventions yielded the strongest empirical support? Which were less effective or inconclusive?

Thank you for this helpful suggestion. We revised Section 6 (Discussion) to include explicit comparisons of effectiveness across intervention types. Cognitive-behavioral and solution-focused models are now identified as having the strongest empirical support, while digital and hybrid models are described as promising but underutilized. Family- and community-based approaches and screening interventions are discussed as context-specific, with variable evidence.

Discussion: Expand on implementation gaps: Elaborate on systemic barriers (e.g., lack of funding, insufficient training for nurses, fragmented health–education coordination) to explain why promising practices are not widely adopted.

Thank you for this suggestion. We expanded Section 6 (Discussion) to address systemic barriers limiting implementation and scalability. The new paragraph cites specific findings from included studies (e.g., Yosep et al., 2022; Yosep et al., 2023; Evgin & Bayat, 2020; Öztürk Çopur & Kubilay, 2021; Kvarme et al., 2016; Hutson & Melnyk, 2022) describing insufficient training, workload and funding constraints, and fragmented coordination between health and education systems. This revision contextualizes why promising practices remain underutilized and identifies strategies, such as protected training time and cross-sector collaboration, to improve implementation and sustainability.

Discussion: Strong points are made about cultural adaptation, but the section could benefit from concrete examples of culturally tailored programs from the reviewed literature. As written, the discussion feels more normative than evidence-driven.

We agree that the section would benefit from stronger empirical grounding. To address this, we revised the Discussion (§6) to include specific examples of culturally tailored interventions drawn directly from the reviewed studies. The revised text now references Brandão Neto et al. (2020), Yosep et al. (2022), Evgin and Bayat (2020), and Kvarme et al. (2016) to illustrate how programs incorporated local pedagogical traditions, family norms, and cultural values to enhance engagement and sustainability. These additions strengthen the evidence base of the section and demonstrate how cultural adaptation operates in practice across diverse contexts.

Discussion: Deepen the digital and equity analysis: Include critical reflection on challenges (data privacy, engagement, unequal access). If possible, highlight case examples of successful culturally tailored or digital interventions. While digital tools are highlighted as promising, the discussion remains surface-level. More analysis is needed on barriers (e.g., digital divide, privacy concerns, ethical delivery) and on why they remain underutilized despite scalability.

Thank you for this helpful comment. We expanded Section 6 (Discussion) immediately following the paragraph on Theme 3 (Digital Interventions) to deepen the analysis of equity, ethics, and implementation challenges. The new text integrates findings from recent systematic reviews and meta-analyses (Batool et al., 2025; Chen et al., 2022; Kamaruddin et al., 2023) to discuss barriers such as the digital divide, privacy concerns, and limited infrastructure, and contrasts these with successful, culturally tailored examples from the reviewed literature (Choi et al., 2021; Hutson and Melnyk, 2022). This revision strengthens the theoretical and practical analysis of why digital approaches remain underutilized despite demonstrated scalability.

Tone down overgeneralizations : Where claims of “consistent” or “strong” evidence are made, ensure they are backed by rigorous findings. Otherwise, qualify with terms like “promising,” “preliminary,” or “context-specific.”

Thank you for this comment. This feedback overlaps with the earlier suggestion to strengthen the critical appraisal of evidence, and both were addressed through the same revisions.

Limitations : Although limitations are addressed in Section 6.1, they appear somewhat perfunctory. For example, the variability in outcomes and methodological rigor could be more closely tied to the implications for interpreting the evidence base.

Thank you for this observation. We substantially revised Section 6.1 (Limitations) to strengthen the critical appraisal of methodological limitations and explicitly connect them to the interpretation of findings. The revised section discusses issues such as small sample sizes, lack of randomization, measurement reliability, and limited generalizability, linking these factors to the strength of the evidence base. It also adds forward-looking recommendations for improving methodological rigor in future studies.

Reviewer 2 Report

Comments and Suggestions for Authors

This manuscript provides a comprehensive synthesis of evidence-based interventions and implementation practices addressing bullying and cyberbullying among youth, with particular attention to the role of healthcare and public health professionals. The integration of the SHIELD framework throughout the analysis is a notable strength, as it provides conceptual coherence and highlights multidimensional approaches (Strengths, Healing, Interventions, Empowerment, Learning, and Development). The manuscript reflects a high degree of interdisciplinary awareness, drawing on psychological, public health, and educational literatures.

The application of the SHIELD framework offers a unifying lens to integrate disparate findings across studies. Clear mapping of interventions to SHIELD components (e.g., Healing with CBT, Empowerment with family engagement, Development with community supports) is well-executed. The review covers multiple intervention types: school-based, therapeutic, digital, family/community-based, and screening/early identification. The attention to cultural responsiveness and equity in the later section adds depth and contemporary relevance. Discussion of how nurses and behavioral health providers can act as “connective tissue” across systems is valuable for practice-oriented readers. Emphasis on early identification, digital access, and interdisciplinary collaboration is well aligned with current public health priorities.

At the same time, the review exhibits certain limitations in clarity, structure, and balance that warrant revision before publication.

  • The introduction needs to be revisited. There's no need for subheadings within the introduction. The introduction can end with research questions without including a "Rationale and Purpose" subheading. Avoid repetition as much as possible. The reason for including healthcare and public health professionals in the research objective should be better explained with references.
  • Conceptual Framework: It focuses on SHIELD, whereas the topic is bullying and cyberbullying. It is necessary to provide a theoretical background on this issue.
  • Recommendations for the discussion
  1. Streamline the narrative: Reduce redundancy by consolidating overlapping discussions of trauma-informed care, digital interventions, and family engagement into tighter thematic syntheses. Certain themes (e.g., trauma-informed care, interdisciplinary collaboration, digital interventions) are repeated across multiple sections, leading to redundancy. Condensing would improve readability.
  2. Strengthen critical appraisal : The review sometimes overstates the effectiveness of interventions without sufficiently qualifying the strength of evidence (e.g., “consistently demonstrated stronger outcomes” may not be justified if based on small, non-randomized samples).
  3. Provide more explicit comparisons of effectiveness across intervention types. For example: Which interventions yielded the strongest empirical support? Which were less effective or inconclusive?
  4. Expand on implementation gaps: Elaborate on systemic barriers (e.g., lack of funding, insufficient training for nurses, fragmented health–education coordination) to explain why promising practices are not widely adopted.
  5. Strong points are made about cultural adaptation, but the section could benefit from concrete examples of culturally tailored programs from the reviewed literature. As written, the discussion feels more normative than evidence-driven.
  6. Deepen the digital and equity analysis: Include critical reflection on challenges (data privacy, engagement, unequal access). If possible, highlight case examples of successful culturally tailored or digital interventions. While digital tools are highlighted as promising, the discussion remains surface-level. More analysis is needed on barriers (e.g., digital divide, privacy concerns, ethical delivery) and on why they remain underutilized despite scalability.
  7. Tone down overgeneralizations : Where claims of “consistent” or “strong” evidence are made, ensure they are backed by rigorous findings. Otherwise, qualify with terms like “promising,” “preliminary,” or “context-specific.”

  1. Limitations : Although limitations are addressed in Section 6.1, they appear somewhat perfunctory. For example, the variability in outcomes and methodological rigor could be more closely tied to the implications for interpreting the evidence base.

The manuscript shows strong potential and addresses a highly relevant public health issue with a well-chosen conceptual framework. However, the narrative would benefit from streamlining, greater critical analysis, and clearer methodological transparency before it is suitable for publication.

Author Response

We are grateful for the reviewers’ thorough and constructive feedback. The manuscript has been substantially revised to improve clarity, balance, and critical analysis. Key revisions include:

  • Removal of unnecessary subheadings in the Introduction.
  • Integration of multiple theoretical frameworks within the Conceptual Framework section.
  • Condensation of redundant narrative elements.
  • More explicit cross-comparisons of intervention effectiveness.
  • Strengthened discussion of systemic and equity-focused implications.
  • Expanded and contextualized Limitations section.

We believe these revisions have significantly improved the manuscript’s clarity, rigor, and contribution to the field and we appreciate the opportunity to improve our submission. 

Reviewer 2 Comments

Author Response

(The authors gave the same response as above.)

Round 2

Reviewer 1 Report

Comments and Suggestions for Authors

No more comments to the authors.

Reviewer 2 Report

Comments and Suggestions for Authors

I appreciate the authors for their thorough revisions. All proposed revisions have been implemented:

1. The introduction has been updated to incorporate the rationale and purpose.

2. The conceptual framework has been revised based on theoretical advancements.

3. The discussion has been reorganized and updated with additional references.

4. Several overlooked limitations have been identified.

The manuscript is now suitable for publication.

In this paper, a systematic review of 12 studies from 2013 to 2023 reveals potential contributions of healthcare and public health professionals to early identification, prevention, and resilience-building. Using the SHIELD framework, six key themes were identified: screening and early identification protocols, family and community involvement, mental health focus, school-based interventions, cognitive-behavioral strategies, and the role of digital platforms. The findings suggest that health professionals can implement trauma-informed and culturally responsive strategies and recommend cross-sector collaboration and equity-centered practices to enhance prevention and support for youth.